# META-OPTIMIZING ML MODEL TRAINING

## ABSTRACT

A major challenge in training large-scale machine learning models is *configuring* the training process to maximize model performance, i.e., finding the best training setup from a vast design space. In this work, we unlock a gradient-based approach to this problem. We first introduce an algorithm for efficiently calculating *metagradients*—gradients through model training—at scale. We then introduce a "smooth model training" framework that enables effective optimization using metagradients. With metagradient descent (MGD), we, e.g., greatly improve on existing dataset selection methods and outperform accuracy-degrading data poisoning attacks by an order of magnitude.

## 1 INTRODUCTION

*How should I clean my data? What architecture should I use?* Training machine learning models entails making many design decisions. When making such decisions, typical practice is to exhaustively search over a small set of standard options. For example, we might try a few well-known data cleaning heuristics, construct a grid over a hyperparameters, and choose the options that yield the best models. However, given that this process explores only a small part of the overall design space, it is unlikely that this approach really yields the *optimal* training configuration.

How can we find optimal (or at least, better) training configurations? Well, deciding on a training configuration—or as we will call it, a set of *metaparameters*—is just a high-dimensional optimization problem. The input space of this problem comprises all possible metaparameter choices, including which datapoints to train on, what model architecture to use, and how to initialize model weights. The objective function takes in a set of metaparameters, trains a machine learning model according to those metaparameters, and then returns a target metric evaluated on that model (e.g., test accuracy). From this perspective, any procedure for selecting metaparameters—including the typical practice of grid-searching over standard options—is just an optimization algorithm, whose goal is to maximize the objective function with respect to the (high-dimensional) input.

Given that selecting metaparameters is "just" a high-dimensional optimization problem, a natural tool to consider is the *gradient*. After all, in many contexts, gradients offer a more effective approach to maximizing high-dimensional functions than grid search. Indeed, for a sufficiently "well-behaved" function $f(x)$ with gradient $\nabla f(x)$, we can optimize $f$ by iteratively updating $x$ in the direction of $\nabla f(x)$. This insight suggests a generic recipe for selecting metaparameters: first, make the objective differentiable with respect to the metaparameters; second, update via gradient steps.

Now, the idea of using gradients to search for metaparameters is not new. Indeed, there is a substantial line of work that aims to optimize metaparameters with gradient-based methods (Maclaurin et al., 2015; Liu et al., 2018; Lorraine et al., 2020; Micaelli & Storkey, 2021; Zhang et al., 2021; Chandra et al., 2022; Engstrom et al., 2024) (see Appendix A for an extended account of related work). However, such methods have not managed to scale beyond relatively small settings. This state of affairs prompts our main question:

> *Can we scalably configure model training using gradient-based methods?*

### 1.1 CONTRIBUTIONS

In this work, we answer this question in the affirmative, adding "gradient descent on metaparameters" to the large-scale ML toolkit. Along the way, we will face—and address—two main challenges: first, that existing methods for computing metagradients do not scale; second, that metagradients of

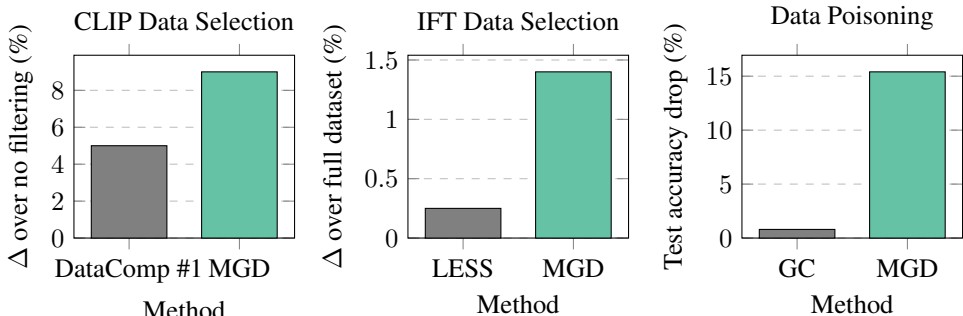

Figure 1: Our proto-algorithm, metagradient descent (MGD), uses gradients to achieve state-of-the-art performance across a variety of applications, including data selection and data poisoning.

standard training routines are not necessarily helpful for optimization (which we show is due to nonsmoothness of the metaparameter optimization landscape).

Addressing these challenges unlocks a simple recipe for solving a broad range of machine learning tasks: (a) frame the task as a continuous optimization problem over metaparameters; (b) design a metasmooth training routine; (c) perform metagradient descent (MGD). Applying this recipe enables state-of-the-art pre-training data selection for CLIP (establishing a new state-of-the-art on DataComp-small and medium (Gadre et al., 2024)); substantially improved instruction-tuning data selection for Gemma-2B (outperforming existing selection methods as well as full-data training); and the first effective *accuracy-degrading* data poisoning attack on deep neural networks (dropping CIFAR accuracy from $92\% \rightarrow 78\%$; previous state-of-the-art only reduces accuracy to $91\%$).

## 2 SCALABLY COMPUTING METAGRADIENTS

In this section we present REPLAY, an algorithm for computing metagradients of large-scale iterative ML algorithms. We first detail the setting, then discuss existing approaches to computing metagradients, and conclude by describing REPLAY.

### 2.1 WHAT IS A METAGRADIENT?

Training a machine learning model is a two-step process. First, we decide on a *training setup*—we must pick, for example, a neural network architecture, a training dataset, and an optimizer for training. Second, we apply the algorithm defined by this training setup to train a model.

Our goal in this paper is to optimize model behavior as a function of the training setup (or, as we call it, the *metaparameters*) using gradients. To this end, we define the following notation:

- Let $\mathbf{z} \in \mathbb{R}^n$ be a vector of continuous metaparameters representing the aspects of training we aim to optimize (e.g., if we want to adjust learning rate/weight decay then $n = 2$). We handle discrete metaparameters (e.g., data selection) through continuous relaxation (e.g., importance weights).

- Let $\mathcal{A}$ be an *algorithm* mapping $\mathbf{z}$ to a trained machine learning model; we assume all other aspects of the training setup outside $\mathbf{z}$ are fixed and thus part of the algorithm $\mathcal{A}$.

- Finally, let $\phi$ be an *output function* mapping a model $\theta$ to a vector $\phi(\theta) \in \mathbb{R}$. For example, $\phi(\theta)$ might represent the validation loss of the model $\theta$. We require that $\phi$ be differentiable with respect to $\theta$, but otherwise make no assumptions on $\phi$.

With this notation in place, we define the *training function* $f := \phi \circ \mathcal{A}$ mapping the training setup $\mathbf{z}$ *directly* to the output function $\phi$ evaluated on the corresponding model.

The *metagradient* is then the gradient of this training function with respect to the metaparameters, $\nabla_{\mathbf{z}} f(\mathbf{z})$. Intuitively, the metagradient is the "direction of steepest ascent" in metaparameter space.

**Our focus: iterative algorithms.** To efficiently compute the metagradient, we restrict our focus to cases where the algorithm $\mathcal{A}$ is *iterative*, i.e., when it can be written in the form

$$\underbrace{\mathcal{A}(z) := \mathbf{s}_T,}_{\text{model state after } T \text{ steps}} \quad \text{where} \quad \underbrace{\mathbf{s}_{t+1} := h_t(\mathbf{s}_t, \mathbf{z}).}_{\text{optimizer step } t} \tag{1}$$

Here, $\mathbf{s}_t$ is the optimizer state at step $t$ (with $\mathbf{s}_0$ being the initial state) and $h_t$ is the *update* mapping from state $t$ to state $t+1$. The form of (1) captures most large-scale training algorithms. For example, if the setup $\mathbf{z} \in \mathbb{R}^T$ is a *per-step* learning rate, and the algorithm $\mathcal{A}$ is full batch gradient descent, then each update $h_t$ is $h_t(\mathbf{s}_t, \mathbf{z}) := \mathbf{s}_t - z_t \nabla \ell(\mathbf{s}_t)$, where $z_t$ is the learning rate at step $t$, $\ell$ is the training loss, and the state $\mathbf{s}_t$ comprises the parameters at step $t$. For more complex algorithms like Adam (Kingma & Ba, 2015), the state $\mathbf{s}_t$ includes terms like gradient moments.

## 2.2 WARMUP: METAGRADIENTS VIA AUTODIFFERENTIATION

A key primitive we leverage to calculate metagradients is *automatic differentiation* (AD)—a standard tool for taking gradients through computer-defined functions. AD takes gradients by decomposing functions into elementary operations with known derivatives, then combining these derivatives using the chain rule. Concretely, AD operates in two passes: a "forward pass," which executes the function of interest and stores intermediate products for each elementary operation; and a "backward pass," which calculates the gradient by propagating chains of partial derivatives using these stored products. For our purposes, we view AD as a black box that calculates the gradient of a many-to-one function (i.e., any $f : \mathbb{R}^d \to \mathbb{R}$) at a given point using only a small constant factor more time than calculating the function itself (plus the space cost of storing the necessary forward-pass products).

**Approach #1: Direct AD.** The direct approach to calculating metagradients exploits the fact that nearly any learning algorithm is itself a sequence of differentiable computer-defined operations—meaning the training function $f$ is *also differentiable*.

Operationalizing this fact to compute metagradients turns out to be challenging. The reason is that AD stores intermediate products for *each* operation, and so the amount of data stored scales with the number of operations in the function of interest. For our training function $f$, this number encompasses *all* the operations used to train a machine learning model. Thus, even in a toy scenario like MNIST training, computing metagradients with naïve AD requires terabytes of memory.

**Approach #2: Exploiting structure with step-wise AD.** A more efficient method for calculating metagradients, *step-wise AD*, leverages the structure of iterative learning algorithms (Werbos, 1990; Maclaurin et al., 2015; Franceschi et al., 2017). Recall from (1) that such algorithms take the form

$$\mathcal{A}(\mathbf{z}) := \mathbf{s}_T, \quad \text{where} \quad \mathbf{s}_{t+1} := h_t(\mathbf{s}_t, \mathbf{z}).$$

Algebraic manipulation (in particular, using the chain rule, the law of the total derivative, and the identity $\mathbf{s}_t = h_{t-1}(\mathbf{s}_{t-1}, \mathbf{z})$) allows us to write the metagradient over an iterative algorithm as

$$\frac{\partial f(\mathbf{z})}{\partial \mathbf{z}} = \frac{\partial \phi(\mathcal{A}(\mathbf{z}))}{\partial \mathbf{z}} = \sum_{t=1}^{T} \frac{\partial \phi(\mathbf{s}_T)}{\partial \mathbf{s}_t} \cdot \frac{\partial h_{t-1}(\mathbf{s}_{t-1}, \mathbf{z})}{\partial \mathbf{z}}. \tag{2}$$

Step-wise AD computes the metagradient by calculating each term in the sum of (2) one at a time. Concretely, the algorithm executes as follows. As a preprocessing step, it trains the model and stores all intermediate states $\mathbf{s}_0, \ldots, \mathbf{s}_T$. Then, the algorithm calculates and sums the terms in (2). It first computes $A_T := \partial \phi(\mathbf{s}_T)/\partial \mathbf{s}_T$, the gradient of the output function $\phi$ with respect to the final state. Then, the algorithm steps through $\mathbf{s}_{T-1}, \ldots, \mathbf{s}_0$ in reverse order, using the previously calculated $\partial \phi(\mathbf{s}_T)/\partial \mathbf{s}_{t+1}$ to calculate (a) the gradient with respect to the state at time $t$ and (b) the gradient with respect to $\mathbf{z}$ at time $t$. Each calculation requires differentiating over only one train step. Finally, the algorithm returns the final metagradient as the sum of the terms.

Despite improving storage overhead compared to "direct AD", step-wise AD is still too space-intensive at scale. After all, this algorithm saves *every* optimizer state.

**Our approach: REPLAY** REPLAY is our algorithm for efficiently and exactly computing metagradients. It uses $\mathcal{O}(k \log_k(T))$ space and requires running the learning algorithm $\mathcal{A}$ a total of $1 + \log_k(T)$ times, with $k$ a user-chosen constant. The main idea is to make the space-intensive subroutine of

step-wise AD—a reverse-order traversal of the optimizer states at each step—much more efficient. After all, step-wise AD stores *all* the states to reverse traverse them. REPLAY modifies step-wise AD to traverse states in less space by exploiting a simple observation: when training is deterministic, one can *reinstantiate* an optimizer state $\mathbf{s}_t$ by "replaying" training from a fixed point $t' < t$—at the compute cost of $t - t'$ training steps. For example, one simple scheme saves every other state, then "replays" the remaining states when (reverse) traversing; this routine stores $T/2$ states but computes an extra $T/2$ model updates compared to storing *all* the states.

REPLAY performs a reverse-order traversal the optimizer states while balancing the compute cost of "replaying" training with the storage cost of saving states. We use a combination of deterministic training (fixing data ordering, data augmentation, and any other randomness in the training process) and an efficient data structure (similar to a segment tree; see Figure 8 in the Appendix) to reverse-order traverse the optimizer states with $\mathcal{O}(k\log_k(T))$ space and an additional $T\log_k(T)$ model steps.

**Remark 1** (Connection to rematerialization)**.** *In a broad sense, both* REPLAY *and step-wise AD can be viewed as special cases of a classical approach in AD called rematerialization (Chaitin et al., 1981; Briggs et al., 1992; Zweig & Padmanabhan, 2000; Griewank & Walther, 2008). To our knowledge, however,* REPLAY *is the first application of this particular rematerialization technique to the problem of computing metagradients through model training.*

**Remark 2** (Reversible learning)**.** *An alternative approach to calculating metagradients that does not save any state is* reversible *learning (Maclaurin et al., 2015), for which one can "invert" previous training states from future ones. We focus here on general (non-reversible) learning algorithms for two reasons: first, even simple algorithms such as SGD without momentum are non-reversible; second, reversibility in practice introduces numerical precision issues.*

## 3 Designing metasmooth training routines

For any training function $f$, REPLAY lets us compute metagradients $\nabla f(\mathbf{z})$ for any setup $\mathbf{z}$. Can we immediately use these metagradients to optimize model training setups? Generally, no: we find that applying REPLAY to a function $f$ representing a standard model training and evaluation routine yields metagradients that are often $\pm\infty$-valued and unhelpful for optimization. Indeed, previous work has observed similar issues optimizing over even (very) small-scale training (Bengio et al., 1994; Maclaurin et al., 2015).

In this section, we show that an underlying source of the issue is the landscape of the metaparameter optimization problem. We then present a framework for modifying standard learning algorithms to admit useful metagradients, i.e., to be *metasmooth*. To use a familiar analogy: just as residual connections and improved initialization schemes improve optimization in standard deep learning algorithms, we introduce an analogous set of modifications to enable optimization with metagradients.

### 3.1 The metaparameter optimization landscape

We first review the notion of smoothness from optimization theory, and then adapt it to the setting of metagradients. The resulting *metasmoothness* metric allows us to quantify (and later, improve) the amenability of the metaparameter optimization problem to gradient-based methods.

**Smoothness.** In optimization theory, the basic property of a function that controls how effectively it can be optimized with first-order methods is *smoothness*. Specifically, a function $f(\mathbf{z})$ is $\beta$-smooth at a point $\mathbf{z}$ if its gradient $\nabla f$ satisfies the property that

$$\|\nabla f(\mathbf{z}) - \nabla f(\mathbf{z}')\| \leq \beta \cdot \|\mathbf{z} - \mathbf{z}'\| \qquad \text{for all } \mathbf{z}', \tag{3}$$

or in other words, if its gradient does not change too quickly around $\mathbf{z}$. To motivate this definition: if a function $f$ is $\beta$-smooth at $\mathbf{z}$, then a step of gradient descent with step size $1/\beta$ will successfully decrease the value of the function (Bubeck, 2014).

**Empirical metasmoothness.** We now adapt the idea of smoothness to metaparameter optimization, with the goal of capturing the spirit of (3) (that gradients at nearby points are similar) while being (a) cheap to compute; (b) easy to interpret; (c) applicable to a learning algorithm $\mathcal{A}$ as a whole rather than a particular training function $f$. To that end, we propose the following definition.

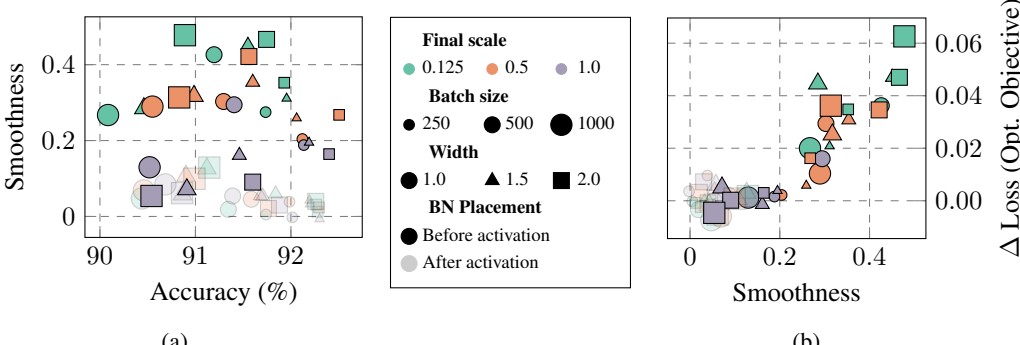

(a)                                           (b)

Figure 2: *(a)* For a variety of training configurations of a ResNet-9 model, we plot metasmoothness (Def. 1) against test accuracy. Increasing width, placing batch normalization before activations, and scaling down network outputs consistently improve metasmoothness, at a minor cost to accuracy. *(b)* Smoother training configurations can be optimized via metagradients more effectively. Here, as in Section 4.2, we use metagradients to gradient ascend on validation loss.

**Definition 1** (Metasmoothness of $\mathcal{A}$). *Fix a small $h > 0$ and a direction $\mathbf{v} \in \mathbb{R}^n$. Let $\mathcal{A}$ be a learning algorithm (Section 2.1) and $\mathbf{z}$ be a metaparameter vector. Let $\mathbf{d} \in \mathbb{R}^d$ be the per-coordinate variation in $\mathcal{A}$, $\mathbf{d} = |\mathcal{A}(\mathbf{z} + 2h\mathbf{v}) - \mathcal{A}(\mathbf{z})|$. The $(h, \mathbf{v})$-metasmoothness of $\mathcal{A}$ at $\mathbf{z}$ is*

$$\widehat{S}_{h,\mathbf{v}}(\mathcal{A}; \mathbf{z}) := \mathrm{sign}(\mathcal{A}(\mathbf{z} + h\mathbf{v}) - \mathcal{A}(\mathbf{z}))^\top \cdot \mathrm{diag}\left(\mathbf{d}/_{\|\mathbf{d}\|_1}\right) \cdot \mathrm{sign}(\mathcal{A}(\mathbf{z} + 2h\mathbf{v}) - \mathcal{A}(\mathbf{z} + h\mathbf{v})). \quad (4)$$

Intuitively, (4) treats each model weight $\mathcal{A}(\mathbf{z})_i$ as its own training function; in particular, notice that $(\mathcal{A}(\mathbf{z} + h\mathbf{v})_i - \mathcal{A}(\mathbf{z})_i)/h$ is exactly the finite-difference approximation of the projection of the gradient of $\mathcal{A}(\mathbf{z})_i$ at $\mathbf{z}$ in the direction of $\mathbf{v}$. Thus, (4) measures the agreement in sign between the gradient at $\mathbf{z}$ and at $\mathbf{z} + h\mathbf{v}$, averages this agreement across weight coordinates, scaling by the variation in each coordinate. Taking a weighted average of sign agreements ensures that $\widehat{S} \in [-1, 1]$, making it easy to interpret. The $\mathrm{diag}(\mathbf{d}/_{\|\mathbf{d}\|_1})$ term weights each agreement proportionally to the scale of the corresponding parameter change (downweighting, e.g., coordinates $i$ that are essentially constant). Finally, observe that Definition 1 is efficient to compute in practice: it requires only three calls to the learning algorithm $\mathcal{A}$.

### 3.2 ESTIMATING AND IMPROVING METASMOOTHNESS

Having established a method for quantifying metasmoothness, we turn to the practical question: how can we design learning algorithms that are amenable to metagradient optimization? To answer this question, we introduce a straightforward framework: given a learning algorithm, explore a fixed menu of possible modifications to the training setup, and choose the combination that maximizes empirical metasmoothness. In practice, we find that this framework allows us to slightly modify learning algorithms in a way that makes them amenable to first-order methods.

As a case study, we study training ResNet-9 on CIFAR-10 (Krizhevsky, 2009). We let the metaparameters $\mathbf{z}$ be a perturbation to the pixels of 1000 random training images ($\mathbf{z} \in \mathbb{R}^{1000 \times 32 \times 32 \times 3}$), and estimate the empirical metasmoothness of different learning algorithms $\mathcal{A}$ at $\mathbf{z} = \mathbf{0}$ using Definition 1. See Appendix D.1 for details of the estimation procedure.

**Metasmooth learning algorithms.** We estimate the metasmoothness of learning algorithms induced by different design choices (batch size, network width, BatchNorm placement, gradient scaling), and report the results in Figure 2 (left). On one hand, "standard" learning algorithms (i.e., those designed without metasmoothness in mind) are not metasmooth. On the other hand, our investigation reveals central factors driving metasmoothness. In addition to "standard" hyperparameters such as batch size and network width playing a role, we find that placing Batch Normalization layers *prior* to nonlinearities (instead of after) and scaling the final layer output are both crucial to metasmoothness. See Appendix F for the full training setup.

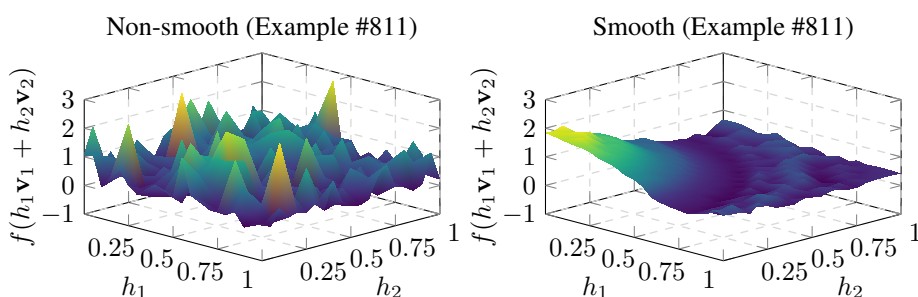

Figure 3: The effect of metasmoothness on the optimization landscape. Each plot visualizes the loss landscape of a (deterministic) learning algorithm $\mathcal{A}$, with the $x$- and $y$-axes representing additive perturbations to 1000 examples in the training set and the $z$-axis representing the resulting model's loss on the test example given in the title. In each row, the left plot is a non-smooth algorithm, and the right plot is a smooth algorithm evaluated on the same example. Overall, empirical metasmoothness seems to strongly correlate with qualitative landscape smoothness. See Figure 10 for more examples.

Finally, in Figure 3, we plot the optimization landscape of both metasmooth (right) and non-metasmooth (left) models. We find that the landscapes of metasmooth models are much smoother and—qualitatively—more straightforward to optimize.

**Metasmoothness/performance tradeoffs?** Figure 2 (left) relates metasmoothness to model accuracy for the considered learning algorithms. While there is no clear trend, the top-performing learning algorithms are not always metasmooth. However, the trade-off is not too severe: the most metasmooth algorithms still achieve near-optimal accuracy. Furthermore, it is possible that with additional searching we could identify even more accurate metasmooth models. Taken together with our previous experiment, our results suggest that jointly searching over metasmoothness and model accuracy is a general recipe for designing learning algorithms that are both performant and metasmooth. Finally, as we mention in our discussion (Appendix B), a fruitful avenue for future work may be to design metasmooth learning algorithms directly, i.e., without heuristics or grid search.

**Does metasmoothness aid downstream optimization?** Recall that our motivation for studying metasmoothness is to develop learning algorithms that we can optimize via metagradient descent. We started with the notion of $\beta$-smoothness from optimization theory, and we adapted it to the setting of metagradients by making some approximations and modifications. Does our final notion of metasmoothness actually predict the utility of metagradients for optimization? Figure 2 (right) demonstrates that metasmoothness strongly predicts our ability to optimize the metaparameters of a given learning algorithm. We use metagradients (computed by REPLAY) to gradient ascend on validation loss with respect to the metaparameters $\mathbf{z}$, and measure the change in model loss.

## 4  CASE STUDIES: OPTIMIZING DATA SELECTION AND TRAINING PIXELS

We now show how to use metagradients to optimize high-dimensional aspects of the training setup, focusing on *optimizing training data*. In Section 4.1, we optimize *selection* of training data, achieving state-of-the-art results on standard data selection benchmarks. In Section 4.2, we optimize the *content* of training data, achieving the first effective accuracy-degrading data poisoning attack on deep neural networks. In each setting we follow the same recipe: frame the task as an optimization problem, modify the learning algorithm of interest to be *smooth*, then solve by first-order optimizing with meta-gradients—which we refer to, in a catch-all manner, as metagradient descent (MGD).

### 4.1  SELECTING TRAINING DATA

Curating a training dataset from a mass of unfiltered data is a necessary and influential step in any large-scale machine learning pipeline. Deciding how to curate such a dataset is a challenging problem that has attracted substantial recent interest (Fang et al., 2022; Abbas et al., 2023; Engstrom et al., 2024; Gadre et al., 2024). In this section, we frame data selection as an optimization problem, and then solve this problem by first-order optimizing with metagradients. We apply our method

to the DataComp benchmark (Gadre et al., 2024) and to the LESS benchmark (Xia et al., 2024) improving on the state-of-the-art in both settings—on DataComp-small, the gap between MGD and the previous state-of-the-art is roughly the same as the gap between the previous state-of-the-art and training on random data.

**Setup.** The goal of dataset selection is to choose a training data subset (out of a broad pool of data) that maximizes trained machine learning model performance. This goal has a natural interpretation as a combinatorial metaparameter optimization problem. In particular, in the language of Section 2.1, for a training set of size $n$, let:

(a) the metaparameters $\mathbf{c} \in \mathcal{C} := \mathbb{Z}_{\geq 0}^n$ be non-negative data counts representing the number of times each training sample repeats in the training data;

(b) the algorithm $\mathcal{A}$ be a standard large-scale learning procedure, which runs on a training set comprising $c_i$ copies of each sample $i$ for $i \in [n]$;

(c) the output function $\phi$ be the loss of the trained model on a target distribution $D$.

Then, defining $f(\mathbf{c}) := \phi(\mathcal{A}(\mathbf{c}))$ (as in Section 2.1), our goal is to find the data counts $\mathbf{c}^*$ that solve

$$\mathbf{c}^* := \arg\min_{\mathbf{c} \in \mathcal{C}} f(\mathbf{c}). \tag{5}$$

**Algorithm.** Metagradients let us *directly* minimize the target task loss (5) with respect to the choice of training data. At a high level, our algorithm operates as follows: we start with a randomly chosen set of training data, then iteratively update the dataset selection using metagradients with respect to importance weights placed on each training datapoint. The specifics of our method are in Algorithm 3; the core ideas are (a) to compute the metagradient using a surrogate algorithm (taking in continuous importance weights rather than integer counts), and (b) to use a signed coordinate descent algorithm to update the training data, ensuring that the data weights stay integer-valued. See Appendix E.1 for details on our precise optimization procedure.

**Evaluation.** We evaluate our data selection algorithm using DataComp Gadre et al. (2024), a standardized framework for evaluating data selection methods for multimodal models, and LESS Xia et al. (2024), an instruction-tuning benchmark for evaluating dataset selection methods for language models. See Appendix E.5 and E.7 for details on the settings.

**Results (DataComp).** MGD greatly outperforms the current state-of-the-art: on DataComp-small, the difference in accuracy between MGD and the current best method is roughly as large as the difference between the previous state-of-the-art (EcoDatum (EcoDatum, 2024)) and training on randomly chosen data (cf. Figure 4). Inspecting scores over the course of the optimization in Figure 4, we find that only a few steps are necessary to outperform previous methods. MGD also establishes a new state-of-the-art on DataComp-medium, attaining an average score by 1.5 percentage points higher than the previous state-of-the-art—see Appendix Table 3 for the numerical results.

**Results (Instruction-tuning).** Comparing with two baselines—training on *all* the data and training with data selected with LESS (Xia et al., 2024)—MGD yields strictly better training dataset selections for each target task (cf. Figure 5). MGD improves most on BBH, a reasoning task, compared to the best baseline (+1.5% accuracy). On MMLU, a knowledge-based task, we outperform baselines by slightly less compared to the best baseline (+0.8%); one explanation is that selecting IFT data lends more control over reasoning than over intrinsic knowledge available in the LM.

## 4.2 ACCURACY-DEGRADING (HUBER) DATA POISONING

The goal of an accuracy-degrading *data poisoning* attack is to degrade the performance of a machine learning model by corrupting a small fraction of its training data. Here, the considered threat model is as follows. The attacker is given a training set $\mathbf{X} = \{x_1, \ldots, x_n\}$ drawn from a distribution $P$, and a function $\theta(\cdot)$ mapping training data to model parameters (representing the learning algorithm used by the victim). The attacker's goal is to return a new training set $\mathbf{X}'$ that differs from $\mathbf{X}$ in at most $\varepsilon \cdot n$ datapoints while inducing model parameters $\theta(\mathbf{X}')$ that perform as poorly as possible on a freshly drawn test set $T$ from $P$.

For large-scale machine learning models, finding strong adversaries has proven challenging—standard loss-minimizing learning algorithms seem quite robust to maliciously-inserted data (Lu

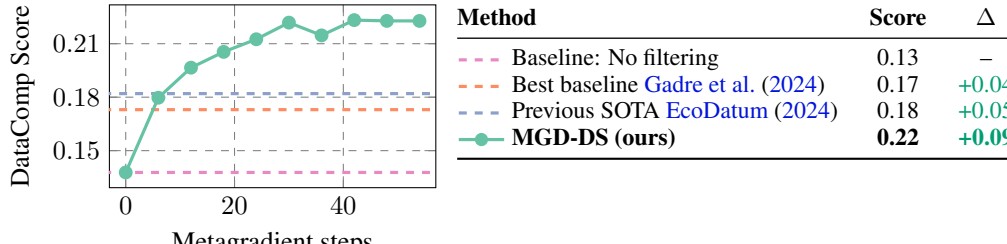

Figure 4: MGD dataset selection greatly outperforms existing methods (improving over the previous SOTA by as much as the previous SOTA improves over no filtering at all). We compare DataComp scores for MGD (over optimization steps), training on the entire candidate pool, the best baseline originally proposed by DataComp, and the previous SOTA (EcoDatum, 2024). Analogous results for DataComp-medium are in Appendix Table 3.

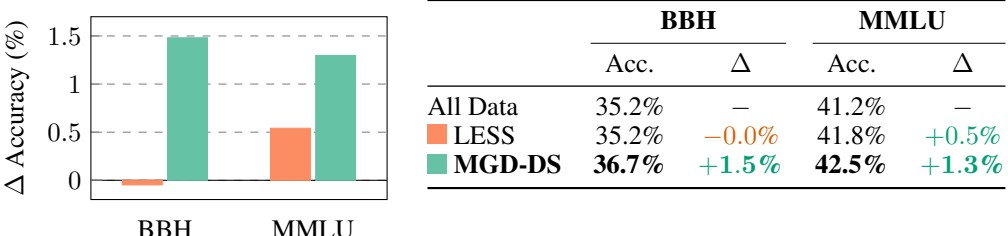

Figure 5: MGD dataset selection outperforms baselines. Comparing to training on all the data: it achieves over double the margin of improvement of LESS on MMLU, and improves by $+1.5\%$ on BBH (where LESS does not improve at all). The $\Delta$ column denotes improvement over not filtering.

et al., 2023). In fact, even constructing attacks that degrade the overall performance of a learning algorithm by more than the adversarial budget $\varepsilon$ is challenging.

**Setup.** We frame data poisoning as a continuous optimization problem to which we can directly apply our metagradient framework. In particular, given a (randomly shuffled) training set $\mathbf{X}$ and validation set $\mathbf{X}_{val}$, we set up the following metaparameter optimization problem (see Section 2.1):

(a) the metaparameter $\mathbf{z} \in \mathcal{X}^{n_p}$ is a tensor of $n_p = \lfloor \varepsilon n \rfloor$ poisoned samples;

(b) the algorithm $\mathcal{A}$ maps metaparameters $\mathbf{z}$ to a trained model $\mathcal{A}(\mathbf{z})$ by replacing the first $n_p$ samples in $\mathbf{X}$ with the samples in $\mathbf{z}$ and then training on the resulting dataset;

(c) the output function $\phi$ evaluates average loss on the validation set $\mathbf{X}_{val}$.

**Algorithm.** To apply our first-order methods to this problem, we start by initializing the poisoned data to be exactly the first $n_p$ samples in $\mathbf{X}$, $\mathbf{z}^{(0)} := \{\widetilde{x}_i^{(0)} = x_i : i \in [n_p]\}$. Then, for $t = 1, \ldots, T$, we sample a minibatch $\mathbf{X}_{val}^{(t)}$ from $\mathbf{X}_{val}$ and use REPLAY to compute the metagradient

$$\mathbf{g}_t = \tfrac{d}{d\mathbf{z}} \left( \sum_{x \in \mathbf{X}_{val}^{(t)}} \ell(x; \mathcal{A}(\mathbf{z}^{(t-1)})) \right),$$

and update the poisoned data using (projected) gradient ascent:

$$\mathbf{z}^{(t)} = \Pi_{\mathcal{X}} \left( \mathbf{z}^{(t-1)} + \eta \cdot \text{sign}(\mathbf{g}_t) \right),$$

where $\Pi_{\mathcal{X}}$ is the projection operator onto the sample space $\mathcal{X}$ (e.g., when $\mathcal{X}$ is the space of image-label pairs, $\Pi_{\mathcal{X}}$ clips pixel values to $[0, 1]$ and ensures labels are valid probability distributions).

**Evaluation.** We use the CIFAR-10 dataset which consists of 60,000 total images each labeled as one of 10 classes. We partition the data into 40,000 training examples, 10,000 validation examples, and 10,000 test examples. We consider a simple 12-epoch CIFAR-10 training procedure, which reaches 92.4% accuracy on the CIFAR-10 test set when applied to the 40,000 training examples. See Appendix F for training hyperparameters.

| deer | horse | frog | deer | frog | horse | airplane | deer | dog | ship |

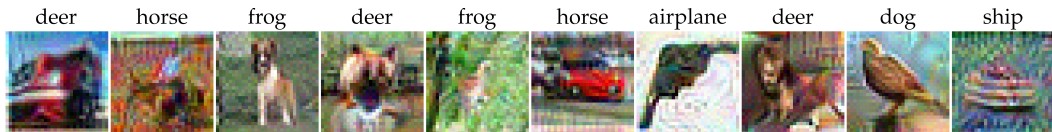

Figure 6: Examples of poisoned images from Section 4.2.

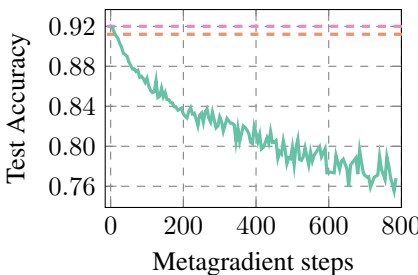

| Model | Acc. | Δ |
|---|---|---|
| - - Original model | 92.0% | − |
| - - GradCancel (Lu et al., 2023) | 91.2% | −0.80% |
| **— MGD-DP (ours)** | **78.1%** | **−13.9%** |
| 1-layer NN (ref.) (Coates et al., 2011) | 83.3% | −8.7% |

Figure 7: For each iteration of MGD ($x$-axis), we train a new model from random initialization on a randomly shuffled training set with the current iterate of poisoned data injected. We evaluate the test accuracy ($y$-axis), and use REPLAY to compute the metagradient. MGD outperforms the best known attack (Lu et al., 2023) by an order of magnitude and (for reference) yields a model with the same accuracy as a single-layer neural network trained on random image features (Coates et al., 2011).

As described above, we allow the adversary to modify (in-place) a fixed, $\varepsilon$-fraction of the training data (in our case, 2.5%) subject to the constraint that the poisoned images still lay in the valid (normalized) image range of $[0, 1]$. We compare our approach—direct optimization of the data poisoning objective using metagradients—to the state-of-the-art "Gradient Cancelling" (GradCancel) method of Lu et al. (2023) (see Appendix F for an explanation of the method).

**Results.** We find that metagradients enable state-of-the-art data poisoning attacks, degrading accuracy by 14%. In particular, when allowed to corrupt 1000 of the 40,000 training samples (2.5%), our method reduces test set accuracy to 78%—for reference, the accuracy of a single-layer neural networked trained on the unmodified CIFAR-10 training set is 83%. The strongest existing data poisoning attack, GradCancel, only reduces test set accuracy by less than 1%.[1] In Figure 6, we visualize the poisoned images and labels found by our method. In Figure 7, we visualize the minibatch loss at each step of the optimization process.

**Remark 3** (Poisoning non-smooth learning algorithms). *Recall that to apply metagradient descent, we alter the learning algorithm $\mathcal{A}$ to be metasmooth (see Section 3.1). This involves making modifications such as switching out max pooling layers for average pooling layers or moving batch normalization layers before activations. How much does the efficacy of our method depend on this smoothness? After all, in practice the adversary cannot control the learning algorithm. To answer this question, we take the poison samples generated by MGD and insert them into the training set of a corresponding standard (i.e., non-metasmooth) learning algorithm. We find that our method still significantly degrades the performance of the model, from $92.8\%$ to $82.6\%$ (a drop of $10.2\%$).*

## 5 CONCLUSION

In this work we add metagradients to the large-scale machine learning toolkit. To do so, we overcome two challenges: (a) calculating metagradients at scale and (b) modifying learning algorithms to be metasmooth—i.e., to admit metagradients that locally predict model behavior. We then successfully calculate and apply metagradients for large-scale models (up to 2B parameters) to select data for CLIP pretraining and instruction fine-tuning, and to poison training data (decreasing overall model accuracy). In Appendix B, we discuss limitations and potential applications of metagradients. Broadly, we are excited to see what metagradients enable beyond optimizing training data.

---

[1]Lu et al. (2023) report a larger drop; this is because we constrain poisoned data to be valid images.

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

# A EXTENDED RELATED WORK

We overview previous work on calculating and applying meta-gradients.

## A.1 CALCULATING METAGRADIENTS

Previous work estimates the metagradient for large-scale models via one of two broad families of methods: implicit differentiation and automatic (explicit) differentiation. Note that in previous literature, synonyms for metagradient include "hyper-gradient" and "outer gradient."

**Implicit differentiation.** One family of methods aims to *approximate* the metagradient. To illustrate the idea behind such approaches, suppose that the learning algorithm $\mathcal{A}$ returns a model state $\theta$ that minimizes a strongly convex loss function $\mathcal{L}(z, \theta)$. Here, the implicit function theorem tells us that

$$\nabla_z f(z) = \overbrace{\left( \frac{d\phi}{d\theta} \bigg|_{\theta = \mathcal{A}(z)} \right)}^{\substack{1 \times p \text{ gradient of output} \\ \text{wrt. final params}}} \underbrace{\left( \frac{\partial^2 \mathcal{L}(z, \theta)}{\partial \theta^2} \bigg|_{\theta = \mathcal{A}(z)} \right)^{-1}}_{\substack{p \times p \text{ inverse Hessian of loss} \\ \text{wrt. final params}}} \overbrace{\left( \frac{\partial^2 \mathcal{L}(z, \theta)}{\partial \theta \, \partial z} \bigg|_{\theta = \mathcal{A}(z)} \right)}^{\substack{p \times n \text{ Jacobian of loss gradient} \\ \text{wrt. metaparameters}}}. \tag{6}$$

The form of (6) yields efficient and accurate estimators for metagradients of models learned by minimizing a strongly convex loss (Bertrand et al., 2020; 2022; Kolter et al., 2020; Blondel et al., 2022; Scieur et al., 2022). Such approaches can extend to estimate metagradients of large-scale, non-convex learning algorithms (Bengio, 2000; Koh & Liang, 2017; Rajeswaran et al., 2019; Finn et al., 2017; Lorraine et al., 2020; Chen & Hsieh, 2020; Bae et al., 2022), but lose any correctness guarantees. Indeed, applying this class of methods in large-scale settings is challenging as doing so requires (a) assuming conditions on the learning algorithm (e.g., Hessian invertibility, continuous differentiability) and (b) efficiently approximating the inverse Hessian (in practice, typically at the cost of estimate accuracy). Finally, implicit function-based approaches are fundamentally limited in that they can only differentiate with respect to metaparameters expressed in the loss function (e.g., these methods can differentiate with respect to the weight decay, but not learning rate).

**Automatic (explicit) differentiation.** Beyond implicit differentiation approaches, there is a long line of work on directly calculating metagradients with AD (see Section 2). Previous work has used AD to estimate metagradients of learning algorithms ranging from those with convex objectives to small neural networks (Hara et al., 2019; Maclaurin et al., 2015; Franceschi et al., 2017; Micaelli & Storkey, 2021; Zhang et al., 2021; Chandra et al., 2022; Scieur et al., 2022). As detailed in Section 2, the primary challenge with (reverse-mode) AD-based approaches to meta-differentiation is storing the intermediate products required for the backward pass. To circumvent this challenge, previous work either (a) only considers settings that are small enough that is possible to differentiate while requiring space that is linear in the number of iterations (i.e., 2 layer networks on MNIST), (b) uses forward-mode AD (Franceschi et al., 2017; Micaelli & Storkey, 2021; Chandra et al., 2022) (which requires no extra storage at the cost of additional compute that scales linearly with metaparameter dimension), (c) only *approximates* the metagradient by calculating over only a few training steps (Liu et al., 2018; Chen & Hsieh, 2020; Finn et al., 2017), or uses (d) a reversible learning algorithm (Maclaurin et al., 2015). The fourth category is a promising direction for reducing space requirements when computing large-scale metagradients, but current approaches require (a) representing model parameters in a fixed-precision format (which current large-scale learning algorithms do not support) in addition to restricting the algorithm to be reversible (e.g., SGD and standard GD do not qualify). A common thread is that algorithms computing metagradients with AD often suffer from numerical instability and overflow issues (Micaelli & Storkey, 2021; Scieur et al., 2022). In relation to previous work on AD, REPLAY (Section 2) can be seen as a strategy for choosing gradient checkpointing (Chaitin et al., 1981; Briggs et al., 1992; Zweig & Padmanabhan, 2000; Griewank & Walther, 2008; Chen et al., 2016) locations in the compute graph (an NP-complete task in general (Naumann, 2008)).

## A.2 Applying metagradients

Previous work applies metagradients to optimize training setup, including distillation (Maclaurin et al., 2015; Lorraine et al., 2020), training data selection (Hara et al., 2019; Engstrom et al., 2024), meta-learning (Finn et al., 2017; Rajeswaran et al., 2019; Hospedales et al., 2021), learning rate/weight decay selection (Micaelli & Storkey, 2021; Chandra et al., 2022), tuning data augmentation (Lorraine et al., 2020), and architecture search (Maclaurin et al., 2015; Liu et al., 2018; Zhang et al., 2021). Beyond optimizing metagradients, methods in data attribution apply metagradients to (Taylor) estimate the effect of dropping training data on model predictions (Koh & Liang, 2017; Grosse et al., 2023; Park et al., 2023). Previous works either (a) calculate metagradients directly with AD (made feasible by working in a very small-scale learning setting) or (b) estimate the metagradient with an implicit function-based approach.

# B  DISCUSSION

In this section, we first present the main limitations of our method and outline future directions.

**Limitations.** Although REPLAY is more efficient than existing methods at computing metagradients, it is still non-trivially more expensive than simply training a model once. The main reason is that metagradients require making a *backwards pass over a backwards pass*. This operation necessarily requires 2-3 times the operations of a backwards pass; furthermore, our current implementation requires `float32`/`tensorfloat32` operations. Finally, standard training operations are often made more efficient by specialized software (e.g., via FlashAttention (Dao et al., 2022)); no such software (yet) exists for backwards-over-backwards operations. Beyond computational issues, successfully applying metagradients requires smooth model training.

**Metasmoothness: connections and future directions.** While Section 3 describes a general procedure for finding metasmooth learning algorithms, an important future direction is to further explore and understand metasmoothness. This includes, for example: (a) characterizing the relationship between metasmoothness and numerical stability (and potentially using techniques from the latter to improve the former); (b) devising improved optimizers and/or architectures that lead directly to metasmooth learning algorithms (akin to skip connections or stable initialization in architecture design); (c) formalizing connections between metasmoothness and other optimization-related phenomena in deep learning (Leclerc & Madry, 2020; Cohen et al., 2022). A related but separate direction is to explore the possibility of using techniques from non-smooth optimization (Clarke, 1990) to perform metagradient descent on non-metasmooth learning algorithms.

**Applying metagradients.** Our methods apply to any ML task that requires optimizing with respect to a metaparameter. These include: poisoning data (generated or simply hosted on the internet) so that it cannot be trained on without permission (i.e., by maximizing training loss with respect to the text); selecting better training data at various stages of the model training lifecycle; and designing better model training routines and architectures with first-order methods. Another direction of future work lies in mitigating the computational limitations of our algorithm. Both (a) small-scale proxy-models (Hoffmann et al., 2022; Engstrom et al., 2024) and (b) low-hanging engineering improvements can likely make calculating metagradients much more efficient.

## C  CALCULATING METAGRADIENTS WITH REPLAY

This appendix contains supplementary material for Section 2. We describe two algorithms in detail: step-wise AD, and our own algorithm REPLAY. Refer to Section 2 for the notation used in this appendix.

### C.1  WARMUP: STEP-WISE AD

We fully describe step-wise AD in Algorithm 1. The algorithm requires storing all $T$ optimizer states, but requires constant memory overhead for each AD call (as each AD call is over a single step), making it feasible to compute for small setups.

---

**Algorithm 1:** metagradients in $\mathcal{O}(T)$ space.

1  `// Store each optimizer state on disk`
2  $\{s_i\}_{i=0}^{T} \leftarrow$ Train model via $A(z)$
3
4  `// Variables; shorthand for` $\frac{\partial f(z)}{\partial z}$ `and` $\frac{\partial f(z)}{\partial s_T}$
5  $\bar{z} \leftarrow 0$
6  $\bar{s}_T \leftarrow \frac{\partial g(s_T)}{\partial s_T}$    `// One reverse-mode AD call`
7
8  `// Reverse-mode differentiate step-by-step`
9  **for** $s_i \leftarrow s_{T-1}$ **to** $s_0$ **do**
10  |    `// One reverse-mode AD call.  Left:` $\nabla_{s_i} f$. `Right:`
        `contribution to` $\nabla_z f$ `at` $i$.
11  |   $\bar{s}_i \leftarrow \bar{s}_{i+1} \cdot \frac{\partial h_i(s_i,z)}{\partial s_i}, \qquad \bar{z}_i \leftarrow \bar{s}_{i+1} \cdot \frac{\partial h_i(s_i,z)}{\partial z}$
12  |
13  |   $\bar{z} \leftarrow \bar{z} + \bar{z}_i$    `// Accumulate metagradient`
14
15  **Return** $\bar{z}$

---

### C.2  REPLAY

We now describe REPLAY, our method for calculating metagradients. For a free parameter $k \in \mathbb{N}$, REPLAY requires storing $\mathcal{O}(k \log_k(T))$ optimizer states and an additional $\mathcal{O}(\log_k(T))$ factor of computation. The free parameter $k$ controls the trade-off between storage and required compute. We fully describe REPLAY in Algorithm 2. REPLAY modifies Algorithm 1 by retrieving the optimizer states in reverse order using a $k$-ary tree structure in lieu of a list of all the stored states.

#### C.2.1  LAZY $k$-ARY TREE

We now describe the $k$-ary tree structure that underlies REPLAY; for a visual reference of this tree with $k = 2$, see Figure 8. For ease of analysis we parameterize the total number of states as $n = T + 1$ (and therefore take $n - 1$ total training steps) when describing this data structure, and assume WLOG that $n$ is an integer power of $k$. At a high level, traversing this tree recursively replays retraining to recover all the optimizer states in reverse order, while deleting states that are no longer needed. We call this tree "lazy" because it retrains only when required to obtain states that are not yet retrieved.

The tree is a complete $k$-ary tree with $n$ leaves (and therefore $\log_k(n)$ depth) structured as follows. We start at the root, then recursively define the rest of the tree. Every node in the tree represents a single optimizer state. The root represents state $s_0$. To recursively define the remaining nodes: each non-leaf node $s_i$ at depth $d$ has $k$ equally spaced (in terms of state number) children starting—from left to right—at state $s_i$ and ending at $s_{i+n/k^{d+1}}$. This means that the leaves correspond—from left to right—to the states $s_0, s_1, \ldots, s_{n-1}$.

We reduce the problem of iterating over the states in reverse to the problem of reverse in-order traversing this tree and yielding *just* the leaves—this is exactly the states in reverse order. A reverse

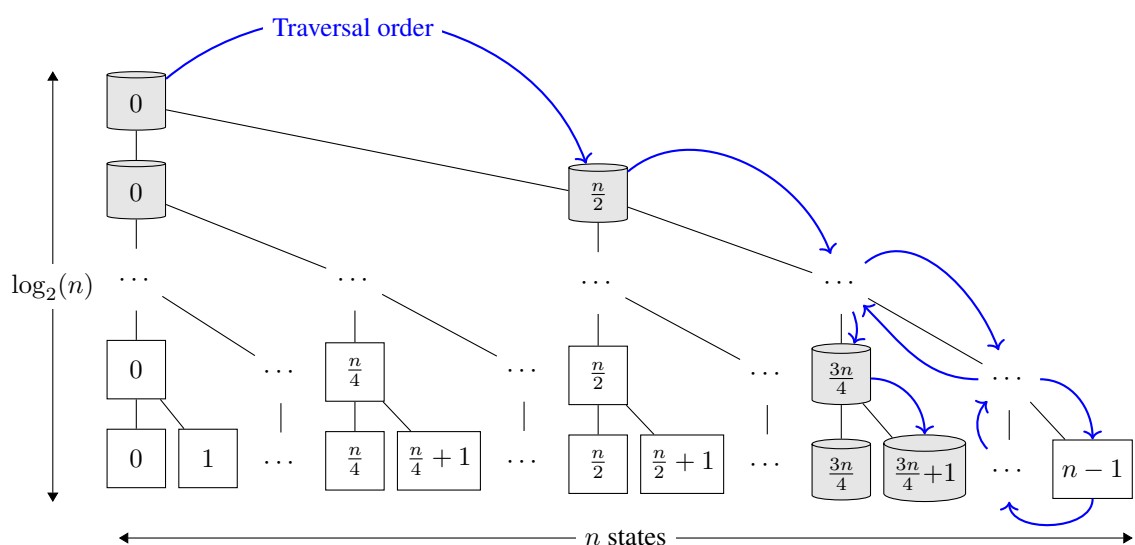

Figure 8: The lazy $k$-ary tree structure for traversing optimizer states in reverse order, with $k = 2$. Recall that $n$ is the number of states (parameterized such that $n = T + 1$). Each node represents the correspondingly numbered state. We give an example of the traversal using the blue arrows in the figure, which denote the traversal path up to state $s_{\frac{3n}{4}+1}$. The gray cylinders 🗄 indicate the states that are stored when the traversal is at state $s_{\frac{3n}{4}+1}$; the other states are not stored at this point in the traversal. Traversing this structure requires storing $\mathcal{O}(\log(n))$ state and computing $\mathcal{O}(n\log(n))$ optimizer steps—compared to $n$ for simply training.

in-order traversal for this $k$-ary tree requires repeatedly: recursively traversing child nodes from largest to smallest, then visiting the parent node. We design the specifics of this traversal to maximize space and compute efficiency. To access the children of a parent node at traversal time, we replay model training from the smallest child state (which is stored in the parent state) to the largest child state and store all the children. We perform this operation recursively each time we traverse a node. After traversing the node's left side (i.e., after ascending from this node), we delete all its child states.

Reverse in-order traversing this tree requires storing at most $k\log_k(n)$ optimizer states at a time, and in aggregate requires retraining the model $\log_k(n)$ times. The argument for each is straightforward. Storage: the traversal requires storing at most $k$ states for each level that it descends (we store $k$ states whenever we first traverse to a parent node) and we remove $k$ states for each level that the traversal ascends (we remove $k$ states after we are done with the left traversal of a parent). Compute: we replay training to reinstantiate the children of every parent node a single time. The $k^d$ parent nodes at level $d$ each require replaying $\mathcal{O}(n/k^d)$ states to reinstantiate children. Therefore, in a traversal, each level requires $\mathcal{O}(n)$ ($k^d \cdot n/k^d$) optimizer steps. There are $\log_k(n)$ levels with parent nodes, which means a total of $\mathcal{O}(n\log_k(n))$ optimizer steps, or a multiplicative factor of $\mathcal{O}(\log_k(n))$ steps compared to model training.

**Algorithm 2:** REPLAY. metagradients in $\mathcal{O}(k\log_k(T))$ space.

```
 1  T ← Lazy k-ary tree for A(z)      // Make lazy k-ary tree of Appendix C.2
 2
 3  // Variables; shorthand for ∂f(z)/∂z and ∂f(z)/∂s_T
 4  z̄ ← 0
 5  s̄_T ← ∂g(s_T)/∂s_T      // One reverse-mode AD call
 6
 7  // Reverse-mode differentiate step-by-step;
       traverse T instead of stored states
 8  for s_i ← s_{T-1} to s_0 ∈ reverse_inorder_traversal(T) do
 9      // One reverse-mode AD call.  Left:  ∇_{s_i} f.  Right:
           contribution to ∇_z f at i.
10      s̄_i ← s̄_{i+1} · ∂h_i(s_i,z)/∂s_i,      z̄_i ← s̄_{i+1} · ∂h_i(s_i,z)/∂z
11
12      z̄ ← z̄ + z̄_i      // Accumulate metagradient
13
14  Return z̄
```

# D  SMOOTH MODEL TRAINING

## D.1  PROCEDURE FOR ESTIMATING SMOOTHNESS

We now describe how we arrive at the smoothness estimates in §3 for a given learning algorithm $\mathcal{A}$ and at a given metaparameter $\mathbf{z}_0$. That is, we sample a random vector $\mathbf{v} \sim \mathcal{N}(0, 1)$ of the same shape as $\mathbf{z}$ and set the small constant $h = 0.1$. For each learning algorithm $\mathcal{A}$, we train three models (one for each of $\mathbf{z} = \mathbf{z}_0$, $\mathbf{z} = \mathbf{z}_0 + h\mathbf{v}$, and $\mathbf{z} = \mathbf{z}_0 + 2h\mathbf{v}$) and use (4) to estimate the average metasmoothness $\overline{S}_{h,\mathbf{v}}(\mathcal{A}; \mathbf{z}_0)$. Concretely, we proceed as follows for each learning algorithm $\mathcal{A}$:

1. Let $\mathbf{z}_0 = \mathbf{0}$ be the metaparameter corresponding to the original dataset.

2. Sample a random perturbation vector $\mathbf{v} \sim \mathcal{N}(0, 1)$.

3. Compute the empirical metasmoothness (4), i.e.,

    (a) Let $\theta_0 := \mathcal{A}(\mathbf{z}_0)$, $\theta_h := \mathcal{A}(\mathbf{z}_0 + h \cdot \mathbf{v})$, and $\theta_{2h} := \mathcal{A}(\mathbf{z}_0 + 2h \cdot \mathbf{v})$ be the model parameters that result from training with training dataset perturbations $\mathbf{z}_0$, $\mathbf{z}_0 + h\mathbf{v}$, and $\mathbf{z}_0 + 2h\mathbf{v}$, respectively.

    (b) Compute the approximate derivatives

    $$\Delta_{\mathcal{A}}(\mathbf{z}_0; \mathbf{v}) = \frac{\theta_h - \theta_0}{h}, \quad \Delta_{\mathcal{A}}(\mathbf{z}_0 + h\mathbf{v}; \mathbf{v}) = \frac{\theta_{2h} - \theta_h}{h}.$$

    (c) Compute the weighting vector $\mathbf{d} = |\theta_{2h} - \theta_0|$

    (d) Compute the average metasmoothness (4), i.e.,

    $$\widehat{S}_{h,\mathbf{v}}(\mathcal{A}; z_0) = \text{sign}(\Delta_{\mathcal{A}}(\mathbf{z}_0 + h\mathbf{v}; \mathbf{v}))^{\top} \cdot \text{diag}\left(\frac{\mathbf{d}}{\|\mathbf{d}\|_1}\right) \cdot \text{sign}(\Delta_{\mathcal{A}}(\mathbf{z}_0; \mathbf{v})).$$

## D.2  OMITTED FIGURES

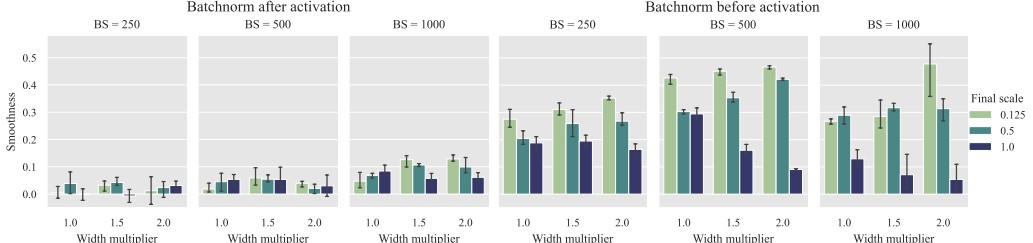

Figure 9: The factors affecting metasmoothness of training a ResNet-9 on the CIFAR-10 dataset. See §3 for details.

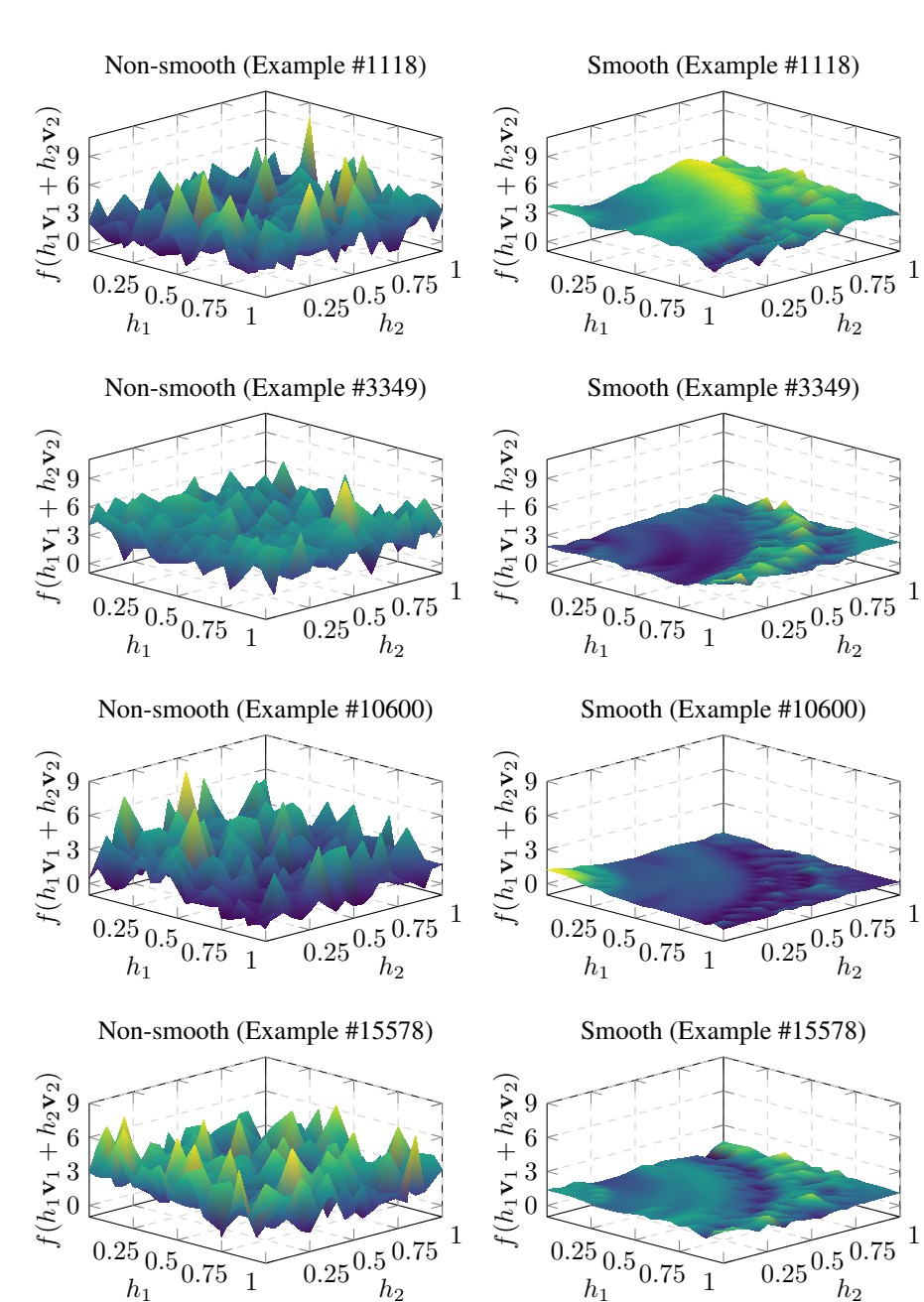

Figure 10: Additional loss landscape visualizations.

# E    METAGRADIENTS FOR DATA SELECTION

This appendix contains pseudocode for the main algorithm used to do dataset selection for DataComp and LESS. It also contains additional implementation details on how metagradients were applied to CLIP, and how they were specifically applied to the DataComp setting.

## E.1    HIGH-LEVEL APPROACH TO DATA SELECTION

We describe the high-level approach to data selection referenced in Section 4.1.

**Idea 1: A surrogate algorithm.** We cannot use metagradients to optimize (5) directly, because the metaparameters of interest $\mathbf{c}$ are discrete counts (and so the algorithm $\mathcal{A}$ is non-differentiable with respect to $\mathbf{c}$). To circumvent this problem, we relax $\mathcal{A}$: we define a surrogate algorithm $\mathcal{A}'_{\mathbf{c}}$ that takes in a *continuous* metaparameter $\mathbf{z} \in \mathbb{R}^n$, whose metagradient we *can* compute, then optimize using the metagradient on $\mathcal{A}'_{\mathbf{c}}$.

This surrogate learning algorithm $\mathcal{A}'_{\mathbf{c}}$ maps a metaparameter $\mathbf{z} \in \mathbb{R}^n$ (representing a perturbation to training data weights) to a machine learning model. The surrogate is defined by a set of counts $\mathbf{c} \in \mathbb{Z}^n_+$, and a hyperparameter $k$ denoting a specific training iteration, both of which we bake into the surrogate algorithm itself. Given a metaparameter $\mathbf{z} \in \mathbb{R}^n$, the algorithm $\mathcal{A}'_{\mathbf{c}}$ trains a model "as usual" using the fixed counts $\mathbf{c}$. That is, it makes $c_i$ copies of each training sample $i$, shuffles and partitions the data into batches, and then at each iteration minimizes the batch loss with a step—just as the original learning algorithm $\mathcal{A}$. At iteration $k$, however, in addition to the original loss on the $k$-th batch, the algorithm upweights *each* training sample $i$ according to the metaparameter $z_i$. In other words, the objective at iteration $t$ of the surrogate algorithm $\mathcal{A}'_{\mathbf{c}}$ is

$$\ell'_t(\theta) := \begin{cases} \sum_{x \in t^{\text{th}} \text{ batch}} \ell(x; \theta) & \text{if } t \neq k \\ \sum_{x \in t^{\text{th}} \text{ batch}} \ell(x; \theta) + \sum_{i=1}^n z_i \ell(x_i; \theta) & \text{if } t = k \end{cases}$$

where $\ell(x; \theta)$ is the training loss on example $x$.

Observe that when $\mathbf{z} = \mathbf{0}_n$, the algorithm $\mathcal{A}'_{\mathbf{c}}$ is identical to the standard learning algorithm $\mathcal{A}$. And while $\mathcal{A}$ was a function of (nondifferentiable) discrete data counts $\mathbf{c}$, $\mathcal{A}'_{\mathbf{c}}$ is differentiable with respect its input $\mathbf{z}$, and so we can compute the metagradient

$$\mathbf{g} := \nabla_{\mathbf{z}} \phi\big(\mathcal{A}'_{\mathbf{c}}(\mathbf{z})\big)\big|_{\mathbf{z}=\mathbf{0}_n}.$$

Intuitively, the entries of the metagradient $\mathbf{g}$ capture the effect of adding an infinitesimal amount of each training sample $i$ to the training data at iteration $k$. A positive entry $g_i$ indicates that adding an infinitesimal amount of sample $i$ to the training data would increase the loss, and a negative entry indicates that adding an infinitesimal amount of sample $i$ to the training data would decrease the loss; the slot at $i$ represents the (estimated) effect of adding a copy of sample $i$ to the training data at every batch containing the sample.

**Idea 2: Block coordinate descent.** We then use the metagradient $\mathbf{g}$ to iteratively update our selected dataset. We update data counts as

$$\mathbf{c} \leftarrow \mathbf{c} - \text{sign}(\mathbf{g}) \odot \mathbf{m}, \quad \mathbf{m} \sim \text{Bernoulli}(p), \tag{7}$$

where $p$ is a hyperparameter controlling the fraction of sample counts to update. This algorithm resembles a block coordinate descent algorithm (Ortega & Rheinboldt, 2000), with the main difference being that we take signed gradient steps with step size 1 (projected onto non-negative integers) to ensure that the counts remain well-defined. As a result, $p$ implicitly controls the algorithm's step size.

Applying (7) concludes a single optimization step. By repeating this process of estimating the metagradient, updating our counts vector, then constructing a new training dataset, we iteratively improve the selected data. Pseudocode for our algorithm can be found in Algorithm 3.

## E.2    FULL SELECTION ALGORITHM

## E.3    DATASET SELECTION USING MGD

When implementing Algorithm 3, there are several differences from the pseudocode below: firstly, rather than selecting $\mathbf{m}$ fully randomly every step, we randomly select a shard comprising fraction

---

**Algorithm 3:** Dataset selection using using metagradient descent (MGD).

---

**Input:** initial data counts $\mathbf{c} \in \mathbb{Z}_{\geq 0}^n$, learning algorithm $\mathcal{A}$, output function $\phi$

**Hyperparameters:** step size $p$, # opt steps $T$, iteration number $k$

1 **for** $t \leftarrow 1$ *to* $T$ **do**

2    $\mathbf{z} \leftarrow \mathbf{0}_n$ // Build input to surrogate

3    $\mathbf{g} \leftarrow \frac{\partial \phi(\mathcal{A}'_{\mathbf{c}}(\mathbf{z}))}{\partial \mathbf{z}}$ // Calculate metagradient using REPLAY

4    $\mathbf{m} \leftarrow$ sample from Bernoulli$(p)$ // Sample indices to step on

5    $\mathbf{c} \leftarrow \mathbf{c} - \text{sign}(\mathbf{g}) \odot \mathbf{m}$ // Take optimization step

6 **Return** $\mathbf{c}$ // Return final data counts

---

$p$ of the data and take steps on all datapoints in the shard (see Section E.4). To mitigate overfitting, we also bake a "minibatch fraction" $q$ into our model output function $\phi$. For example, if $\phi$ calculates model loss on the ImageNet train set, each time $\phi$ is called, we randomly sample fraction $q$ of the ImageNet train set to evaluate on.

**Adapting the CLIP loss function to our surrogate learning algorithm.** Here, we explain how dataweights are incorporated into the CLIP loss function—the formulation given in Section 4.1 is actually slightly simplified and incorrect, as it does not account for cross terms in the CLIP contrastive loss. As a refresher, we first state the "vanilla" CLIP loss function, $\ell$, as it is defined in Radford et al. (2021). Letting $b$ be the batch size and $d$ be the embedding dimension, and $\mathbf{x}$ be the training batch at timestep $k$. Recall that the CLIP model internally has two "submodules": and image embedder, and a text embedder. We then use these to obtain image embeddings $E_I \in \mathbb{R}^{b \times d}$ and text embeddings $E_T \in \mathbb{R}^{b \times d}$ from $\mathbf{x}$. We then compute the image-wise scores, or logits, for this batch as $S = E_I E_T^\top$ [2]. Then, we can define the CLIP loss (as a function of the logits) as

$$L(S) = \frac{1}{2}(L_I(S) + L_T(S)),$$

where $L_I$ and $L_T$ are row-wise and column-wise cross-entropy losses, respectively:

$$L_I(S) = \sum_{i=1}^{b} \log\left(\frac{\exp(S_{i,i})}{\sum_{j=1}^{b} \exp(S_{i,j})}\right), \quad L_T(S) = \sum_{i=1}^{b} \log\left(\frac{\exp(S_{i,i})}{\sum_{j=1}^{b} \exp(S_{j,i})}\right).$$

We now wish to relax $L$ into a new function $L'$ that supports an additional input $\mathbf{z} \in \mathbb{R}^n$, where $\frac{\partial L'}{\partial \mathbf{z}}$ resembles the metagradients with respect to dataweights. In order to do this, we imagine expanding passing the *entire dataset* $D$ into our embedder to obtain $E_I'$ and $E_T'$, and take our new logits $S' = E_I' E_T'^\top \in \mathbb{R}^{n \times n}$.

There are some additional key conditions our relaxation $L'$ should satisfy. Particularly: when $\mathbf{z} = \mathbf{0}_n$, we should recover the normal CLIP loss $L$, and when $\mathbf{z}$ is all 0's except for a single entry $i$, $L'$ should act as if $i$ had been appended to the original batch $\mathbf{x}$. In addition, $L'$ should always have meaningful partials with respect to $\mathbf{z}$, even when some values in $\mathbf{z}$ are 0.

Letting $\mathbf{1}_{i=j}$ and $\mathbf{1}_{i\neq j}$ be indicator variables and letting $\mathbf{1}_k \in \{0,1\}^n$ be the indicator vector for the $k$-th batch, we find that the definition

$$L'(S', \mathbf{z}) = L_I'(S', \mathbf{z}) + L_T'(S', \mathbf{z}),$$

where

$$L_I'(S', \mathbf{z}) = \sum_{i=1}^{n} (z_i + (\mathbf{1}_k)_i) \log\left(\frac{\exp(S_{i,i}')}{\sum_{j=1}^{n} \exp(S_{i,j}')\left(\mathbf{1}_{i=j} + \mathbf{1}_{i\neq j}(z_j + (\mathbf{1}_k)_j)\right)}\right)$$

and

$$L_T'(S', \mathbf{z}) = \sum_{i=1}^{b} (z_i + (\mathbf{1}_k)_i) \log\left(\frac{\exp(S_{i,i}')}{\sum_{j=1}^{n} \exp(S_{j,i}')\left(\mathbf{1}_{i=j} + \mathbf{1}_{i\neq j}(z_j + (\mathbf{1}_k)_j)\right)}\right)$$

---

[2]The CLIP model scales these logits by a temperature parameter $\tau$ before applying the softmax. While we omit $\tau$ in our definitions, it can be easily incorporated. All our experiments use temperature scaling.

satisfy these conditions.

Finally, we let define the loss for the entire batch $\ell'$ as a function of $\mathbf{z}$ and model parameters $\theta$ which outputs the loss calculated according to $L'$ above. To summarize, letting $\mathbf{x}^{(t)}$ denote the $t$-th training batch, the loss function $\ell_t$ at step $t$ of our surrogate learning algorithm $\mathcal{A}'$ for CLIP training is:

$$\ell_t'(\theta) := \begin{cases} \ell(\mathbf{x}^{(t)}; \theta) & \text{if } t \neq k \\ \ell'(\mathbf{z}; \theta) & \text{if } t = k. \end{cases}$$

We find that this empirically works well for obtaining meaningful metagradients with respect to dataweights in the CLIP setting, and yields to strong dataset selection results.

### E.4 SCALING MGD FOR CLIP AND DATACOMP

MGD is highly scalable, allowing it to be applied to large-scale settings like training CLIP models. In particular, computing metagradients is only up to a constant factor more expensive than training a model normally. Here, we outline challenges we faced in scaling MGD in this setting, and how they were resolved. Specifically, we will explain how we efficiently calculated metagradients for CLIP models and efficiently tracked/shuffled our dataset selection from step-to-step despite its large storage footprint.

**Computing metagradients.** Due to the large batch size used in the CLIP contrastive loss, we implement manual gradient checkpointing to make the operations computationally feasible on our hardware. The most memory-intensive operation are model forward passes (and its gradients): obtaining the image and label embeddings given raw pixel data and tokens. So, we manually make gradient checkpoints before this operation, allowing us to run the embedder in minibatches to avoid memory issues. This setup also naturally lends itself to parallelization across multiple GPU's, which we make use of to further speed up our computations.

**Loading, writing, and storing data.** Due to the data-intensive nature of training large models like CLIP and our need to frequently produce new datasets at each optimization step, we found that using the webdataset Webdataset (2024) format given by DataComp was restrictively slow. To circumvent this, we rewrote all data following the format of FFCV Leclerc et al. (2022), allowing us to load and write data much faster. Specifically, we divided the entire candidate pool into 8 base shards. Once we trained a model, we choose one of the 8 shards, compute metagradients corresponding to all datapoints in the shard, take a gradient step on them, and rewrite the shard. This roughly corresponds to $p = \frac{1}{8}$ in Algorithm 3, which we empirically worked well for optimizing. In following steps, we always choose one of the 8 *original* shards to calculate metagradients for—this ensures that points removed from the dataset in some optimization step can return if they have a negative metagradient.

We also observed that always stepping on the sign causes the sizes of the shards to grow over time: stepping based on the sign of the metagradient does not decrease the weight on a positive-weight datapoint if its dataweight is already 0, so our steps are biased towards increasing the size of the shards. To combat this blowup, after some number of optimization steps, we choose a fixed shard size and enforce that subsequent steps must not change the size of the shards—the step size thereafter is controlled by hyperparameter $q$ representing the fraction of datapoints in a shard which are incremented. We experimented both with randomly sampling which points are added or removed, and stepping on the datapoints with the top $q$ and bottom $q$ metagradients; the latter seems to give empirically better performance.

To maintain randomness during shuffling, we implement an 8-way dataloader which would shuffle all 8 shards individually. Then, to sample a batch of $b$ datapoints, we would sample $b/8$ datapoints from each shard and concatenate them to fill our batch. This works better than simply sampling our entire batch from a single shard, as (especially in later optimization steps) shards may contain a high number of duplicate datapoints, which causes CLIP's contrastive loss function to misbehave if they appear in the same batch.

To minimize disk space used, old shards can be deleted once they become "stale". Specifically, if shard $s$ is rewritten into shard $s'$, all future optimization steps will never read $s$ again, and $s$ can safely be deleted. Thus, when running MGD for a large number of steps and potentially rewriting each shard multiple times, the total disk space used by our algorithm is constant in the number of steps we take: it stores the 8 most recently written shards on disk at any given time, and any other shards are deleted to save space.

## E.5   DETAILS PERTAINING TO THE DATACOMP BENCHMARK

DataComp (Gadre et al., 2024) is a multimodal model training competition and benchmark for evaluating dataset selection methods. DataComp provides a *fixed* learning algorithm chosen in advance by the organizers and a large fixed *candidate pool* of internet data. The goal is to choose a subset of the candidate pool—possibly with repeated datapoints—that yields the best-performing model after training with the given learning algorithm, as measured by a predetermined set of 38 benchmarks. Given a submission subset, the mean score on the evaluation datasets for a model trained with that subset is taken as the final "score."

We summarize the DataComp competition here, and we refer readers to the original paper Gadre et al. (2024). DataComp is a framework to compare different training dataset selection techniques. Participants submit a training dataset (which, for our purposes, is a subset of a larger dataset), upon which a CLIP model is trained from scratch with a fixed learning algorithm, model architecture, and number of training steps. We focus on DataComp-small, which has a candidate pool of 12.8 million samples. The number of training steps in this case is also fixed at 12.8 million samples.

We try to match the optimization hyperparameters enforced by DataComp as closely as possible. As a refresher, our ADAM Kingma & Ba (2015) update step can be written as

$$\theta_{t+1} = -\alpha_t \cdot \left( m_t / \left( \sqrt{v_t + \varepsilon_{\text{root}}} + \varepsilon \right) + \lambda \theta_t \right) \tag{8}$$

where $m_t$ and $v_t$ are running estimates of the first and second moments of the gradients, respectively, $\lambda$ represents weight decay, $\alpha$ represents the learning rate, and $\varepsilon$ and $\varepsilon_{\text{root}}$ are hyperparameters to avoid blowup. Our training hyperparameters can be found in Table 1 and are identical to those mandated by DataComp-small, aside from a positive $\varepsilon_{\text{root}}$ added for numerical stability. The values of $\varepsilon_{\text{root}}$ and $k$ (the step at which metagradients are calculated) were chosen to empirically maximize metasmoothness.

Table 1: Hyperparameters for the CLIP DataComp experiments.

| Hyperparameter | Value |
|---|---|
| DataComp Scale | small |
| Model | ViT-B/32 |
| Train compute (MACs) | $9.5 \times 10^{16}$ |
| Pool size | 12.8M |
| # samples seen | 12.8M |
| Batch size | 4096 |
| Training batches | 3125 |
| $k$ | 2800 |
| Learning rate | $5 \times 10^{-4}$ |
| AdamW $\beta_1$ | 0.9 |
| AdamW $\beta_2$ | 0.98 |
| AdamW $\varepsilon_{\text{root}}$ | $1 \times 10^{-17}$ |
| Warmup | 500 |

Our experiments are also run on an incomplete subset of the entire DataComp candidate pool. DataComp did not store the raw image and text files when assembling their dataset; they only stored a list of URL's to download data from. Due to the nature of the internet, for various reasons, some of these URL's no longer point to the same data (or no longer point to any data at all). Thus, after ignoring these broken links, our candidate pool is only around $80\%$ of the size of the original DataComp candidate pool when it was collected in 2023. All our results are obtained by running our methods on this subset of the DataComp pool.

**Evaluation tasks.** In order to ensure that our method is truly improving trained models' performances on the *entire target distribution* and not overfitting to the target set, for each of the 38 evaluation tasks used by DataComp, we attempted to separately create a disjoint target and validation set (DataComp only creates test sets for each task). Thus, metagradients were computed on the target sets and model performance was evaluated on the validation set, before submitting with the official DataComp script and evaluating on the test sets. This ensures that our method's generalization ability is being evaluated, and we are not overfitting to our target set.

For various reasons, creating target splits was not possible for all 38 tasks; we summarize our setup in Table 2.

Table 2: All DataComp evaluation tasks. The "Target set" column refers to whether metagradients were taken on the target set corresponding to this dataset.

| Dataset | Task | Test size | Train size | Val size | Main metric | Target set |
|---|---|---|---|---|---|---|
| Caltech-101 Fei-Fei et al. (2004) | Object recognition | 6085 | 2754 | 306 | mean per class | ✓ |
| CIFAR-10 Krizhevsky (2009) | Visual recognition | 10000 | 45000 | 5000 | accuracy | ✓ |
| CIFAR-100 Krizhevsky (2009) | Visual recognition | 10000 | 45000 | 5000 | accuracy | ✓ |
| CLEVR Counts Johnson et al. (2017); Zhai et al. (2019) | Counting | 15000 | 65000 | 5000 | accuracy | ✓ |
| CLEVR Distance Johnson et al. (2017); Zhai et al. (2019) | Distance prediction | 15000 | 65000 | 5000 | accuracy | ✓ |
| Country211 Radford et al. (2021); Thomee et al. (2016) | Geolocation | 21100 | 37980 | 4220 | accuracy | ✓ |
| DTD Cimpoi et al. (2014) | Texture classification | 1880 | 3384 | 376 | accuracy | ✓ |
| EuroSAT Helber et al. (2019); Zhai et al. (2019) | Satellite imagery recognition | 5400 | 19440 | 2160 | accuracy | ✓ |
| FGVC Aircraft Maji et al. (2013) | Aircraft recognition | 3333 | 6001 | 666 | mean per class | ✓ |
| Food-101 Bossard et al. (2014) | Food recognition | 25250 | 70750 | 5000 | accuracy | ✓ |
| GTSRB Stallkamp et al. (2011) | Traffic sign recognition | 12630 | 35289 | 3920 | accuracy | ✓ |
| ImageNet 1k Deng et al. (2009) | Visual recognition | 50000 | 1276167 | 5000 | accuracy | ✓ |
| ImageNet Sketch Wang et al. (2019) | Visual recognition | 50889 | N/A | N/A | accuracy | * |
| ImageNet V2 Recht et al. (2019) | Visual recognition | 10000 | N/A | N/A | accuracy | * |
| ImageNet-A Hendrycks et al. (2019) | Visual recognition | 7500 | N/A | N/A | accuracy | * |
| ImageNet-O Hendrycks et al. (2019) | Visual recognition | 2000 | N/A | N/A | accuracy | * |
| ImageNet-R Hendrycks et al. (2020) | Visual recognition | 30000 | N/A | N/A | accuracy | * |
| KITTI distance Geiger et al. (2012); Zhai et al. (2019) | Distance prediction | 711 | N/A | N/A | accuracy | † |
| MNIST LeCun (1998) | Digit recognition | 10000 | 55000 | 5000 | accuracy | ✓ |
| ObjectNet Barbu et al. (2019) | Visual recognition | 18574 | N/A | N/A | accuracy | * |
| Oxford Flowers-102 Nilsback & Zisserman (2008) | Flower recognition | 6149 | 1836 | 204 | mean per class | ✓ |
| Oxford-IIIT Pet Parkhi et al. (2012); Zhai et al. (2019) | Pet classification | 3669 | 3312 | 368 | mean per class | ✓ |
| Pascal VOC 2007 Everingham et al. (2010) | Object recognition | 14976 | 14096 | 1566 | accuracy | ✓ |
| PatchCamelyon Veeling et al. (2018); Zhai et al. (2019) | Metastatic tissue cls. | 32768 | 289912 | 5000 | accuracy | ✓ |
| Rendered SST2 Zhai et al. (2019) | Sentiment classification | 1821 | 7013 | 779 | accuracy | ✓ |
| RESISC45 Cheng et al. (2017); Zhai et al. (2019) | Satellite imagery recognition | 6300 | 22680 | 2520 | accuracy | ✓ |
| Stanford Cars Krause et al. (2013) | Vehicle recognition | 8041 | 7329 | 814 | accuracy | ✓ |
| STL-10 Coates et al. (2011) | Visual recognition | 8000 | 4500 | 500 | accuracy | ✓ |
| SUN-397 Xiao et al. (2010) | Scene recognition | 108753 | N/A | N/A | accuracy | ‡ |
| SVHN Netzer et al. (2011); Zhai et al. (2019) | Digit recognition | 26032 | 68257 | 5000 | accuracy | ✓ |
| iWildCam Beery et al. (2021); Koh et al. (2020) | Animal recognition | 42791 | 147084 | 5000 | macro F1 score | ✓ |
| Camelyon17 Bandi et al. (2018); Koh et al. (2020) | Metastatic tissue cls. | 85054 | 365900 | 5000 | accuracy | ✓ |
| FMoW Christie et al. (2018); Koh et al. (2020) | Satellite imagery recognition | 22108 | 103261 | 5000 | worst-region acc. | ✓ |
| Dollar Street Rojas et al. (2022) | Object recognition | 3503 | 13842 | 1537 | worst-income top-5 acc. | ✓ |
| GeoDE Ramaswamy et al. (2024) | Object recognition | 12438 | 44488 | 4943 | worst-region acc. | ✓ |
| Flickr30k Young et al. (2014) | Image and text retrieval | 31014 | N/A | N/A | R@1 | § |
| MSCOCO Lin et al. (2014) | Image and text retrieval | 5000 | N/A | N/A | R@1 | § |
| WinoGAViL Bitton et al. (2022) | Commonsense association | 3563 | N/A | N/A | Jaccard score | § |

---

*No train or val set exists for this dataset, so we were unable to create disjoint target and val sets.

†We were unable to use this dataset due to technical difficulties.

‡Both the train and val sets were used by DataComp to make their test set, so we were unable to create disjoint target and val sets.

§Retrieval tasks were not used for metagradients.

### E.6 DATACOMP-MEDIUM RESULTS

| Method | Score | $\Delta$ |
|---|---|---|
| Baseline: Basic filtering | 0.285 | – |
| Best baseline Gadre et al. (2024) | 0.328 | +0.04 |
| Previous SOTA Wang et al. (2024) | 0.388 | +0.10 |
| **MGD-DS (ours)** | **0.402** | **+0.12** |

Table 3: DataComp-medium results.

### E.7 SELECTING IFT DATA

In this section, we describe the details of the IFT setting of Xia et al. (2024), as well as the details of our method.

**Setting.** The setting contains a fixed data pool: instruction fine-tuning data from a data pool consisting of four combined IFT datasets (cf. Table 5 and Xia et al. (2024) for more information). The goal is to select the data that yields the best possible task performance for a LoRA fine-tuning run. We adapt a LoRA to a Gemma-2B model (the pretraining-only Gemma-2B model) using the LoRA configuration from Xia et al. (2024).

**Data splits.** See Table 4 for a description of the available data for each task, along with the task setup details. Xia et al. (2024) constructed these extra samples by drawing from the ICL samples given in the tasks originally. Note that we drop TydiQA from the original work of Xia et al. (2024) as there are not enough samples to select with (there is only one from each category, for a total of 7).

**Method.** We execute Algorithm 3 with $k$ as 150 steps from the end of training and the Bernoulli parameter $p$ controlling the step size as 0.2. At each step, we choose a "minibatch" with a size equal to half the target set and a quarter of the target set for BBH and MMLU, respectively (that is, we only select to optimize performance on a fraction of the target set at a time). We model select over iterates and hyperparameters by (a) choosing the top three steps in terms of validation loss for each run (b) selecting the best one in terms of full train set accuracy (including the part that we trained on). We perform this procedure—akin to Pareto optimization (Jin & Sendhoff, 2008)—because the validation set is so small (as the overall set of samples is very small) that it is difficult to select models without overfitting otherwise.

We compare with two baselines: training on the full dataset (i.e., training on the entirety of all the data for a single epoch), and LESS (we use the data selected according to "LESS-T" (Xia et al., 2024), following the recommendation of 4 epochs).

For model training, we train with ADAM ($\beta_1 = 0.95, \beta_2 = 0.975$, decoupled weight decay as $10^{-5}$) and a one-cycle linear schedule starting at $10^{-6}$ of the maximum learning rate, reaching the peak over 25% of training, then ending at 0.1 of the maximum learning rate. We insert a positive $\varepsilon_{\text{root}}$ into the inverse square root term in the ADAM update to prevent metagradient (and to a lesser extent update) blowup (see Eq. 8). The model training is the same across selected data, except that we use $\varepsilon_{\text{root}} = 10^{-7}$ for MGD-selected data and $\varepsilon_{\text{root}} = 10^{-9}$ for the other runs (we select the optimal parameter for each class of method). We additionally hyperparameter select for the best learning rate across each baseline by minimizing validation set loss; LESS performs best with a smaller learning rate (0.00024 for BBH and 0.00012 for MMLU) than training on the full dataset or with MGD (0.0006 for both). We normalize the loss of each training sample by taking the mean across predicted tokens during training, and do not divide by the batch size (important for scaling the $\varepsilon_{\text{root}}$ term, but otherwise ADAM is invariant to the scale).

**Selecting smooth model training for MGD.** For MGD runs, we jointly select learning rate and $\varepsilon_{\text{root}}$ using the smoothness metric of Section 3. We find that the choice of $\varepsilon_{\text{root}}$ term is important (just as the choice of $\varepsilon$ is important in standard ADAM training); choosing a much larger term results in non-smooth training. We also find that metagradients are sensitive to learning rate schedule; choosing a much larger or smaller maximum learning rate results in non-smooth training.

Table 4: Overview of datasets used in IFT dataset selection (from Xia et al. (2024)).

| Dataset | # Shot | # Tasks | $n_{\text{target}}$ | $n_{\text{val}}$ | $n_{\text{test}}$ | Answer Type | Type of Task |
|---------|--------|---------|---------------------|------------------|-------------------|-------------|--------------|
| MMLU | 5 | 57 | 57 | 228 | 18,721 | Letter options | Knowledge/Recall |
| BBH | 3 | 23 | 23 | 46 | 920 | COT and answer | Reasoning |

Table 5: Details of IFT training datasets.

| Dataset | # Instance | Sourced from | Prompt Len. | Completion Len. |
|---------|-----------|--------------|-------------|-----------------|
| FLAN V2 | 100,000 | NLP datasets and human-written instructions | 355.7 | 31.2 |
| CoT | 100,000 | NLP datasets and human-written CoTs | 266 | 53.2 |
| Dolly | 15,011 | Human-written from scratch | 118.1 | 91.3 |
| Open Assistant 1 | 55,668 | Human-written from scratch | 34.8 | 212.5 |

### E.8 ADDITIONAL IFT RESULTS

Figure 11: MGD dataset selection improves the validation loss over metagradient steps, demonstrating our method's efficacy. However, the gap between loss on samples MGD directly optimizes on and the validation samples widens over the number of iterates, and there is overfitting depending on the number of steps taken.

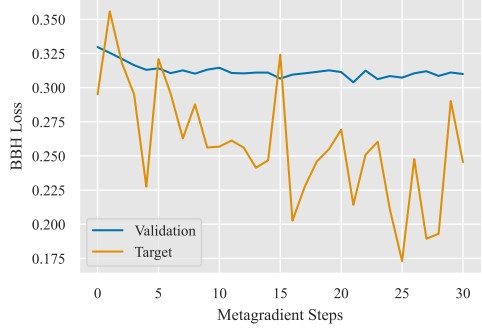 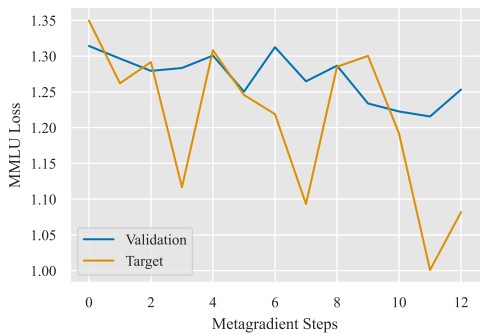

# F  ACCURACY-DEGRADING DATA POISONING

## F.1  BACKGROUND ON GRADCANCEL

We briefly review the Gradient Cancelling attack (Lu et al., 2023) used as a baseline in our experiments. We refer the reader to the original paper for details. Here we highlight the key ideas.

At a high level: Gradient Cancelling (GC) explicitly aims at making a specific malicious parameter configuration reachable through retraining on the poisoned dataset. The attack operates in two phases:

1. **Parameter Generation**: The attacker generates a target malicious model parameter independently, often using a direct parameter corruption method like Gradient-based Parameter Corruption (GradPC) (Lu et al., 2023). The end result of this phase is a target model parameter $\theta_p$ that achieves low accuracy on the test set, but is close to the original parameter $\theta_0$ derived from training on the clean dataset.

2. **Poison Data Crafting**: In the second phase, GC finds values of the poison data that induce a near-zero gradient at the target parameter $\theta_p$. This is achieved by solving a gradient cancellation optimization problem: specifically, GC minimizes the total gradient of the loss function (with respect to the model parameters) evaluated over the combined (clean and poisoned) dataset, aiming to ensure that the gradient at the malicious parameter $\theta_p$ approaches zero.

## F.2  HYPERPARAMETERS

Table 6: Hyperparameters used in the ResNet-9 (Jordan, 2024) CIFAR-10 poisoning experiments. The augmentations used are normalization, random horizontal flip, and random translate (2 pixels)

| Hyperparameter | Value |
| --- | :---: |
| Learning rate | 0.5 |
| $\beta_1$ | 0.85 |
| Weight decay | $10^{-5}$ |
| Exclude BatchNorm | True |
| Optimizer | SGD |
| Batch size | 250 |
| Epochs | 18 |
| Starting learning rate fraction | 0.5 |
| Relative min. learning rate | 10000 |
| Scheduler max. iterations | 50000 |
| Nesterov momentum | True |
| BatchNorm $\varepsilon$ | $10^{-5}$ |
| BatchNorm momentum | 0.5 |
| Final bias | True |
| Width multiplier | 2.0 |
| Final scale | 0.125 |
| Initial scale | 2.0 |
| Batchnorm location | Before activation |
| Activation function | GELU |
| Pooling type | Average |
| Test-time augmentation | True |

