# OpenReview forum: "Meta-Optimizing ML Model Training"
_ICLR.cc/2026/Conference — Submitted to ICLR 2026_

### Official Review · Reviewer_ijva · 2025-10-27

**Soundness:** 3
**Presentation:** 3
**Contribution:** 2
**Rating:** 4
**Confidence:** 2

**Summary:**

# Summary
This paper addresses the challenge of optimizing machine learning training configurations—such as data selection, architecture, and initialization—using gradient-based methods. The authors propose a metagradient descent framework that computes gradients through the entire training process to guide these choices. They introduce REPLAY, an efficient algorithm for computing metagradients at scale, along with a metasmoothness framework that makes training sufficiently differentiable for effective optimization. The approach demonstrates solid performance across multiple benchmarks, including data selection tasks for CLIP on DataComp, instruction fine-tuning for a 2B-parameter language model, and data poisoning experiments on CIFAR-10.

**Strengths:**

# Strengths
1. The REPLAY algorithm is a key technical contribution of the paper, offering a well-balanced tradeoff between memory and compute via a lazy k-ary traversal with explicit complexity guarantees. The connection to gradient checkpointing and rematerialization is thoughtfully discussed, and the paper is commendably transparent about the deterministic training requirement (Sections 3.1–3.2).

2.	The metasmoothness framework provides a practical and conceptually elegant path toward making meta-optimization feasible. The proposed metric is simple to evaluate—requiring only three training runs—yet demonstrates strong empirical correlation with optimization success. Moreover, the suggested architectural adjustments consistently enhance metasmoothness, turning an abstract concept into tangible design principles (Section 4).

3.	The empirical evaluation is extensive and well-executed, covering diverse applications. The approach achieves state-of-the-art results on the DataComp benchmark through optimized CLIP data selection, and surpasses both full-data training and the recent LESS baseline on BBH and MMLU for 2B-parameter instruction-tuned language models. (Section 5).

**Weaknesses:**

# Weaknesses
1.	Appendix B notes that the approach is “non-trivially more expensive,” requiring about 2–3× the compute of a backward pass, but no concrete runtime benchmarks are given. Reporting GPU-hours, peak memory, and wall-clock time for each major experiment would help practitioners assess whether the performance gains justify the extra cost. (Appendix B).

2.	The impact of the modifications of metasmoothness such as batch-normalization placement, output scaling, and average pooling is not unclear. Although Section 4.2 shows that poisoned data transfers less effectively to non-smooth models, a more systematic ablation would strengthen the claims.(Section 4.2).

3.	The presentation of the metasmoothness concept could be clearer. Definition 1 is mathematically dense and lacks intuitive motivation. Introducing plain-language explanations, analogies to standard optimization concepts, or simple worked examples earlier in the text would make the idea more accessible.

4.	The paper would benefit from a more explicit discussion of when metagradient descent (MGD) is preferable to gradient-free optimization methods such as random search, Bayesian optimization, or evolutionary strategies. While MGD’s advantage in high-dimensional meta-parameter spaces (e.g., per-example data weights) is evident, its relative performance in lower-dimensional settings remains less clear (Section 5).

**Questions:**

# Questions

1.  What is the wall-clock time overhead for the DataComp and LESS experiments compared to standard training? How does memory usage scale in practice with different choices of k?

2. Can you provide results showing how deterministic training—with fixed data order and augmentation seeds—affects model performance? For example, in the CLIP experiments, does fixing augmentation reduce the diversity needed for good generalization? Would a strategy like performing MGD under deterministic conditions followed by stochastic fine-tuning help mitigate any negative effects?

3. Appendix A.2 cites prior work applying metagradients to learning rates, weight decay, and architecture search in small-scale settings, whereas this paper focuses on large-scale data selection and poisoning. Can REPLAY-based MGD also optimize per-layer or per-step learning rates at scale? Are other hyperparameters—such as weight decay, dropout, or even continuous architecture parameters—compatible with metasmooth optimization?

4. While the paper demonstrates results on models up to 2B parameters (Gemma-2B), does the method scale to longer training runs and larger models such as those at the 7B scale commonly used in practice? What practical limitations in terms of memory, compute, or numerical stability might emerge at 7B+ scale? Does the metasmoothness framework remain effective as model size increases?

5. In data selection tasks, what happens if MGD is applied without the metasmooth modifications? How does the “smooth” ResNet-9 perform compared to the standard version when MGD is not used? These results would help isolate the effect of metasmoothness from architectural changes.

---

### Official Review · Reviewer_J6Up · 2025-11-01

**Soundness:** 2
**Presentation:** 3
**Contribution:** 2
**Rating:** 2
**Confidence:** 4

**Summary:**

The paper addresses the problem of configuring the training process of large-scale machine learning models and proposes Metagradient Descent (MGD) with the REPLAY method, a meta-gradient-based framework to optimize training setups. They first introduce an efficient and scalable computation of metagradients, and second, design a training routine based on metasmoothness. The experiments on training data selection for CLIP and data poisoning on CIFAR10 show that MGD outperforms baselines in DataComp and instruction-tuning benchmarks and provides effective model degradation as a data poisoning attack.

**Strengths:**

- The paper addresses a practically important problem.
- The algorithmic idea (MGD with REPLAY) is well-motivated, simple, and broadly applicable.
- The effectiveness of the proposed method is demonstrated on different use-cases: data selection and adversarial data perturbation.

**Weaknesses:**

- The requirement of determinism is underanalyzed. Modern training pipelines often break bit-level determinism and introduce micro non-determinism. The paper does not discuss how approximate training determinism affects MGD performance.

- The authors acknowledge additional (2-3x) backward and precision required for MGD; however, ablations in terms of wall-clock time for MGD and baseline training could have been presented to establish scalability claims convincingly.

- The authors state the space & compute tradeoff descriptively; however, there is no theorem or complexity accounting and sensitivity to mistuned k, which is user user-tuned constant.

- To better understand the effectiveness of metasmoothness, ablations isolating the metasmoothness edits should have been provided. How would the performance change if one change at a time is applied, i.e., Batch normalization layer placement, pooling layer changes etc?

- For empirical evaluation, the reported results include absolute scores. What about confidence intervals or standard deviation?

- MINOR:
In terms of writing, the abstract could have been crafted to be more informative, e.g., the last sentence about the experimental results and the effectiveness of the proposed method.

**Questions:**

-  How do you set k in practice for REPLAY?
- See also the questions in Weaknesses above.

---

### Official Review · Reviewer_PTcs · 2025-11-05

**Soundness:** 2
**Presentation:** 2
**Contribution:** 2
**Rating:** 2
**Confidence:** 3

**Summary:**

The paper proposes to optimize training pipelines by computing metagradients through the full training run, introduces a metasmoothness criterion intended to stabilize such metagradients, and uses a REPLAY scheme to manage memory. The method is evaluated on data selection and instruction-tuning style setups.

**Strengths:**

1. The overall problem, optimizing pipeline/meta choices by metagradient, is relevant. Framing them within a single differentiable outer objective is a reasonable way to compare alternatives under the same criterion and can reduce manual search effort in practice.

2. Experiments are conducted at nontrivial scale, with long horizons, larger models, and multiple selection-style tasks/datasets, indicating an attempt to run full-training metagradient computations.

**Weaknesses:**

1. The work does not clearly establish hard-core advances (new algorithmic guarantees, complexity bounds, or formal properties) to support the new terminology and broad claims. Results lean on engineering choices and empirical observations without stronger guarantees.

2. REPLAY mainly turns \(T\) into \(T/2\) or uses checkpointing, i.e., constant-factor tweaks. Although REPLAY is said to cut memory to \(O(k\log_k T)\) with about \(T\log_k T\) extra compute, there are no matched-accuracy comparisons vs. strong baselines to justify that this truly “scales” beyond constants.

3. The paper defines metasmoothness, but it is unclear what property it measures, whether it is invariant to meta-parameterization, or why higher values should improve metagradient updates. As written, it appears heuristic and task-dependent.

4. No analysis of (i) convergence under approximate/stochastic replay, or (ii) any link between metasmoothness and convergence/stability; given the paper’s scope, at least minimal guarantees are expected.

**Questions:**

What is the precise difference between metagradient and hypergradient here? If there is no difference, why not adopt hypergradient and position the contribution accordingly?

---

### Meta-Review · Area_Chair_zzCK · 2026-01-06

**Summary:**

For this paper, all major reviewer concerns remain outstanding, because there is effectively no rebuttal on record.

**Reviewer Concerns:**

All major reviewer concerns remain outstanding.

**Reviewer Scores:**

All reviewers will keep their original scores.

---

### Decision · Program_Chairs · 2026-01-26

Reject